# Comparison of Two Low-Dose Regimens of Intravenous Fentanyl for Pain Relief During Labor: A Double-Blind Randomized Controlled Trial

**DOI:** 10.3390/healthcare13172236

**Published:** 2025-09-07

**Authors:** Veeraphol Srinil, Panjai Inphum, Sukanya Srinil

**Affiliations:** 1Department of Obstetrics and Gynecology, Khon Kaen Hospital, Khon Kaen 40000, Thailand; veeraphol.sri@cpird.in.th; 2Department of Anesthesiology, Khon Kaen Hospital, Khon Kaen 40000, Thailand; panjai.inp@cpird.in.th

**Keywords:** labor pain, fentanyl, analgesia, pain measurement, obstetric, pregnancy, opioid

## Abstract

**Background:** Concerns exist regarding the lowest effective dose of opioids in opioid-naïve pregnancies. This study aimed to compare the effectiveness of 25 µg vs. 50 µg fentanyl in relieving labor pain. **Methods:** In total, 122 term-singleton pregnant females, who planned vaginal delivery, were randomized to receive 25 µg or 50 µg intravenous fentanyl, followed by hourly doses—as needed—for labor pain relief. The primary outcome was the comparison of pain score reduction 30 min after treatment between these regimens. Secondary outcomes included maternal and neonatal safety, total fentanyl dose administered, maternal satisfaction with the fentanyl dosing regimen, and breastfeeding, which were analyzed using appropriate statistical tests. **Results:** Within-group analysis revealed significant pain score reduction 30 min after fentanyl injection: −1.57 (95% confidence interval, CI −2.1 to −1.1, *p* < 0.001) and −1.69 (95% CI −2.2 to −1.2, *p* < 0.001) for 25 µg and 50 µg fentanyl groups, respectively. No significant differences in the pain reduction were observed in between-group comparisons (0.3, 95% CI −0.6 to 1.2, *p* > 0.999), including secondary maternal and neonatal outcomes. Total fentanyl dose was significantly lower in the 25 µg group compared with the 50 µg group (32.8 ± 13.3 vs. 60.2 ± 22.1, *p* < 0.001). **Conclusions:** A 25 µg intravenous fentanyl dose can reduce VAS score, used for evaluating labor pain 30 min after treatment, and is comparable to a 50 µg intravenous fentanyl dose. Given the efficacy of the reduced dosage of fentanyl, this study suggests using 25 µg intravenous fentanyl as an alternative initial dosing for labor pain relief.

## 1. Introduction

Labor is a natural process involving the gradual onset of painful uterine contractions that increase in frequency and intensity until childbirth [1]. Severe labor pain is attributed as a primary reason for women requesting cesarean sections [2]. Therefore, effective labor pain management is essential. Epidural analgesia is the standard method for alleviating labor pain; however, it requires anesthesiologists and teams to ensure patient safety [3]. Given the need for specialized expertise in administering epidural analgesia, many healthcare providers have resorted to alternative pain-relief methods. Parenteral opioids such as pethidine, fentanyl, remifentanil, butorphanol, and nalbuphine are commonly used in these cases [4].

Pethidine is a traditional opioid used to treat labor pain; however, its active metabolite can depress neonatal respiration, which limits its use [5]. Advances in the management of severe acute pain recommend the use of potent and rapid-onset opioids, such as remifentanil and fentanyl. These opioids have a shorter duration of action, and remifentanil, in particular, does not produce active metabolites that may contribute to adverse effects [6]. Fentanyl is primarily metabolized to norfentanyl, an inactive metabolite, while its minor metabolites remain insufficiently studied regarding clinical activity [7,8]. Consequently, both remifentanil and fentanyl have been widely used in labor pain management, demonstrating effective analgesia with generally acceptable maternal and neonatal safety profiles [9]. Although remifentanil is recommended for intravenous patient-controlled analgesia [10], its routine use remains limited in many middle-income countries, particularly within obstetric practice.

Fentanyl is a 4-anilidopiperidine synthetic opioid that binds to mu (µ)-opioid receptors, inhibiting nerve activity and reducing pain perception [11]. Its onset of action occurs immediately after intravenous injection, and its duration of action is 30–60 min [4]. Fentanyl is used in various medical fields and is available worldwide [12]. Research supports its effectiveness in relieving labor pain, with evidence covering different doses, regimens, and routes of administration [13,14,15,16,17,18]. Intermittent intravenous dosing is commonly used to mitigate concerns about cumulative effects, with 50–100 micrograms (µg) administered every 1–2 h [13,14,15,16,17,19]. Variations in dosing protocols have also been reported. One study employed an initial loading dose of 25 µg every 5 min for four consecutive doses, followed by 25 µg every 20 min on patient request, with a maximum cumulative dose of 500 µg [18]. Another study used a simpler regimen of 25 µg administered hourly for two doses [15].

Although fentanyl does not generate clinically significant active metabolites, reported neonatal adverse effects—such as abnormal intrapartum fetal monitoring, low Apgar scores, the need for neonatal resuscitation, and administration of naloxone after bolus doses of at least 50—suggest that dosing strategy and timing, rather than metabolism, may underlie these risks [13,14,16]. Thus, concerns exist regarding the initiation of the lowest effective opioid dosage in opioid-naïve pregnant women and the potential adverse neonatal outcomes. Moreover, a systematic review was unable to determine the most appropriate dosage regimen or opioid type with the least side effects [9,20].

Therefore, this study aimed to evaluate and compare the effectiveness of two low-dose fentanyl regimens (25 µg and 50 µg) for relieving pain during the active phase of labor.

## 2. Materials and Methods

### 2.1. Study Design

This double-blind, parallel-group, randomized, controlled trial was conducted at a tertiary hospital that manages approximately 4200 childbirths annually and serves as a referral center for both medical and obstetric complications, managing approximately 250 high-risk pregnancies each month. Participants were recruited between 16 June 2023 and 3 August 2024. Research assistant nurses in the labor ward obtained written informed consent from all eligible participants during the latent phase of labor, when pain was typically absent or mild. This approach upheld ethical standards of respect, voluntariness, and adequate decision-making time; therefore, women presenting in the active phase of labor were not enrolled.

This study was approved by the Institute Review Board of Human Research on 30 March 2023 (approval number: KEF66006) and was registered in the Thai Clinical Trials Registry (TCTR) on 30 May 2023 (identification number 20230530005). This study adhered to the ethical principles outlined in the Declaration of Helsinki for research involving human participants and followed the CONSORT 2010 guidelines for clinical trials [21].

### 2.2. Participants and Recruitment

Eligible participants were pregnant women aged 18 years or older who were admitted for planned vaginal delivery and requested analgesia during the active phase of labor. The inclusion criteria were singleton pregnancy, cephalic presentation, gestational age between 37 + 0 and 42 + 0 weeks, normal intrapartum cardiotocography upon admission, cervical dilation between 5 and 8 cm [13,22], and a visual analog scale (VAS) score ≥ 4 [23,24].

The exclusion criteria included having received opioids within the previous 24 h; a respiratory rate of 10 or fewer breaths per minute; bradycardia (heart rate less than 60 bpm); oxygen saturation from pulse oximetry below 95%; a diagnosis of severe bronchial asthma, glaucoma, or heart or liver disease; allergies or previous adverse reactions to opioids; opioid dependence within the past year; use of antidepressants within the last 14 days; cognitive impairments; and intellectual disabilities.

Participants were withdrawn from this study if the primary outcome (VAS score) could not be collected. This occurred when the participants delivered before receiving the allocated intervention or delivered less than 30 min after fentanyl administration.

### 2.3. Randomization and Blinding

The randomization process involved a computer-generated sequence—created by a statistician—based on the block randomization method, with randomly selected block sizes of two, four, and six in a 1:1 ratio. Treatment allocations were concealed in sequentially numbered, opaque, sealed envelopes, which were opened by the attending nurse when participants requested analgesia and reported at least moderate pain (VAS ≥ 4) [24,25].

Blinding was maintained for participants, attending nurses, obstetricians, outcome assessors, statisticians, and investigators. For each participant, a set of 10 identical pharmacist-prepared syringes containing the assigned fentanyl dose was provided in sealed code envelopes. To maintain blinding, the syringes were prepared by pharmacists who were not involved in patient care. All additional doses were drawn from these identical syringes, ensuring the same concentration as the initial dose. Any unused syringes were returned to the pharmacy for accountability.

### 2.4. Data Collection

Baseline demographic and clinical variables (age, body weight, body mass index (BMI), parity, cervical dilation, and vital signs) were retrieved from electronic medical records.

Outcome measures were collected prospectively by trained research nurses. Maternal pain was assessed at the peak of uterine contractions using the VAS. The pain scores were recorded immediately before starting the protocol and 30 min after fentanyl administration. Four sets of observations were documented at baseline and at 30, 60, and 120 min after the intervention, if only one dose was administered. For participants who received multiple doses, observations were recorded before and 30 min after each fentanyl injection. All initial, additional, and rescue doses were systematically recorded to ensure accurate assessment of drug utilization.

Sedation was assessed with the Pasero Sedation Scale (SS) [26]. Maternal safety outcomes—including respiratory rate, blood pressure, pulse rate, and oxygen saturation (SpO_2_)—were recorded at baseline (immediately before the intervention) and at 30, 60, and 120 min post-treatment. Episodes of nausea or vomiting were also documented. Neonatal outcomes included intrapartum fetal monitoring, Apgar score, need for resuscitation, naloxone use, and nursery/neonatal intensive care unit (NICU) admission. Postpartum questionnaires were administered 2 h after delivery to evaluate breastfeeding initiation and maternal satisfaction with the study treatment.

### 2.5. Procedures

At enrollment, all eligibility criteria were verified. Pain was assessed with the VAS immediately after the uterine contractions subsided, and cervical dilation was confirmed by obstetricians or nurses to determine the stage of labor. Eligible women who requested analgesia were then randomly assigned to one of the two treatment groups.

Participants were randomized to receive either 25 µg or 50 µg of fentanyl intravenously. The intervention group received 2 mL containing 25 µg of fentanyl, while the control group received 2 mL containing 50 µg of fentanyl. All doses were prepared with 0.9% normal saline and administered intravenously over 1–2 min.

Additional doses according to the assigned regimen (25 µg vs. 50 µg) were administered intravenously every h upon the participant’s request. For participants who continued to experience moderate to severe pain (VAS ≥ 4) 30 min after receiving the initial regimen, a rescue dose of 25 µg intravenous fentanyl was offered. To ensure safety, the maximum cumulative dose was limited to 500 µg per treatment course [18]. Nurses, who were blinded to treatment allocation, calculated cumulative use assuming 50 µg per dose and alerted the investigators when the maximum was approached. If the maximum dose was reached, intramuscular tramadol 100 mg was planned as an alternative analgesic. Metoclopramide (10 mg) was administered intravenously if nausea or vomiting occurred. In addition to the fentanyl protocol, all participants received the same standard obstetric management during the active phase of labor.

Accordingly, fentanyl was not a routine treatment for relieving labor pain, and there were concerns regarding the new treatment protocol in the local context. Continuous intrapartum cardiotocography was performed electronically for 20 min after fentanyl injection. Abnormal fetal monitoring included minimal variability, repetitive late and variable decelerations, acute bradycardia, and prolonged decelerations. Conservative management was opted for in these cases, including the left lateral decubitus position, intravenous infusion of isotonic saline solution, and oxygen therapy until a normal trace was observed.

### 2.6. Outcomes

The primary outcome was the difference in mean VAS pain score reduction 30 min after fentanyl treatment between the 25 µg and 50 µg regimens.

Secondary outcomes were categorized into maternal and neonatal safety outcomes. Maternal outcomes included total and additional doses of fentanyl, weight-normalized value (µg/kg), mode of birth, duration of labor, breastfeeding initiation, satisfaction with the study treatment (not related to the results of the childbirth), and adverse effects of fentanyl (sedation, vital signs, nausea/vomiting). Neonatal outcomes included intrapartum fetal monitoring, Apgar score ≤ 7 at 5 min, need for resuscitation, administration of naloxone, and admission to the nursery or NICU.

### 2.7. Sample Size Calculation

The sample size was calculated based on the primary outcome of pain reduction on the VAS, 30 min after treatment. The minimum clinically significant difference (MCSD) in pain relief was set at 0.9 cm on the VAS, based on previous studies, where a reduction of approximately 0.9–1.6 cm was considered meaningful [13,16,17,27,28]. A pilot study was conducted to estimate the mean and standard deviation of VAS scores for the 25 µg and 50 µg fentanyl groups, yielding values of 7.29 ± 2.04 and 7.01 ± 1.83, respectively, with a pooled variance of 1.94.

Using a two-sided test with α = 0.05 and power = 80%, the required sample size to detect a 0.9 cm difference was 110 participants. To account for potential dropouts (estimated at 5%), the final target sample size was 122 participants.

### 2.8. Statistical Analysis

Data were analyzed according to the intention-to-treat principle. The primary outcome was compared between the two fentanyl regimens using the Mann–Whitney U test. A generalized estimating equation (GEE) population-averaged model was used to estimate the effect size, which was defined as the mean difference in pain score reduction with a 95% confidence interval (CI) across different time points. The Bonferroni correction was applied to adjust for multiple comparisons at these time points. Because the participants gave birth before the scheduled observation times, missing pain score data were assumed to be missing at random. These were handled using all available individual data, accounting for correlations in the GEE-repeated measurements.

Fentanyl dosing was recalculated as a weight-normalized value (µg/kg), defined as the total amount administered—calculated as the sum of protocol doses (number of doses × assigned dose) plus any rescue doses—divided by body weight. Because body weight and BMI data were not normally distributed, potential outliers were assessed using the interquartile range (IQR) method, with outliers defined as values exceeding 1.5 × IQR above the upper quartile.

Continuous data were tested for normality, presented as the mean ± standard deviation (SD) when normally distributed, and compared using an independent *t*-test. Data with a non-normal distribution were presented as the median and interquartile range (IQR) and compared using the Mann–Whitney U test. Categorical data are presented as frequencies and percentages and were analyzed using the Pearson chi-square test. A *p*-value < 0.05 was considered statistically significant. Statistical analyses were performed using Stata software version 18.0 (StataCorp, College Station, TX, USA).

### 2.9. Patient and Public Involvement

The participants were not involved in the design, outcome measurements, or interpretation of the results. The study findings will be shared with participants and the public through a scientific article.

## 3. Results

A total of 2065 women with singleton, cephalic, live, and term fetuses who were planning vaginal delivery were screened for eligibility. Of these, 132 participants met the inclusion criteria and were randomized into two treatment groups, as depicted in the CONSORT flow diagram (Figure 1). Participants were withdrawn from this study if the primary outcome (VAS score) could not be collected: six women in the 50 µg group and four in the 25 µg group delivered before receiving the allocated intervention, and one woman in each group delivered less than 30 min after fentanyl administration.

Baseline characteristics showed no statistically significant differences between the two groups (Table 1). The heaviest participant weighed 102.9 kg in the 25 µg group and 90.5 kg in the 50 µg group, both below the calculated outlier thresholds (109.7 kg and 99.9 kg, respectively); thus, no true outliers were identified. Approximately 80% of the participants in both groups requested pain relief when cervical dilation reached 5–6 cm. The median baseline VAS scores were 8.9 (IQR: 7.5, 10) and 9.2 (IQR: 8, 10) for the 25 µg and 50 µg fentanyl groups, respectively. In total, 122 participants were allocated to receive the intervention (61 per group). However, pain score data at 30 min post-treatment were available for 120 participants (60 per group), and these data were used in the primary outcome analysis.

Table 2 demonstrates that pain scores significantly decreased from baseline to 30 min after fentanyl treatment, with mean changes of −1.57 (95% CI −2.1 to −1.1) and −1.69 (95% −2.2 to −1.2) in the 25 µg and 50 µg fentanyl groups, respectively. Both reductions exceeded the minimum clinically significant difference (MCSD) for labor pain relief of 0.9 cm, indicating meaningful pain reduction in both groups.

The mean pain scores in both groups increased from 30 min to 60 min after treatment, with mean changes of 1.5 (95% CI 0.8 to 2.1) in the 25 µg fentanyl group and 1.3 (95% CI 0.5 to 2.0) in the 50 µg fentanyl group. Multiple comparisons of mean differences at baseline, 60 min, and 120 min within each fentanyl group (25 µg and 50 µg) showed no statistically significant differences (*p* > 0.999). Although VAS scores were slightly lower in the 25 µg group at all time points, the difference remained small (0.3 cm at 30 min after treatment), and no statistically significant differences in pain reduction were observed between the two fentanyl groups (*p* > 0.999) (Table 2).

Pain grade scales between 25 µg and 50 µg fentanyl regimens also demonstrated no statistically significant differences (*p* = 0.219) (Table 3). The linear prediction graph of the VAS pain score at the four time points for both groups is presented in Figure 2.

Table 4 shows that the total fentanyl dose was significantly lower in the 25 µg group compared with the 50 µg group (mean ± SD; 32.8 ± 13.3 vs. 60.2 ± 22.1, *p* < 0.001). Seventeen participants in the 25 µg group and nine in the 50 µg group required more than one protocol dose. Among these, overweight participants accounted for two and three cases, and obese participants for five and two cases, respectively. The majority were of normal weight (10/17 in the 25 µg group and 4/9 in the 50 µg group). Rescue doses were requested by five women, only one of whom was obese. No statistically significant differences were observed between groups regarding the need for additional doses (23.7 vs. 19.7%, *p* = 0.201).

The cumulative fentanyl dose in both groups remained well below the study protocol maximum of 500 µg. In the 25 µg group, total doses ranged from 25 to 75 µg, with a mean ± SD of 32.8 ± 13.3 µg. In the 50 µg group, the range was 50 to 150 µg, with a mean ± SD of 60.2 ± 22.1 µg. No participant approached the maximum limit. Maternal and neonatal outcomes did not differ significantly between the 25 µg and 50 µg intravenous fentanyl groups (Table 5). Five participants (8.2%) in the 25 µg fentanyl group and seven participants (11.5%) in the 50 µg fentanyl group experienced a change in Pasero SS from “awake” to “occasionally drowsy” (*p* > 0.999). However, none of the participants demonstrated desaturation (SpO_2_ ≤ 94%) [29], hypotension, or syncope. Twenty participants (32.8%) in the 25 µg fentanyl group and twenty-three participants (37.7%) in the 50 µg fentanyl group demonstrated abnormal intrapartum fetal monitoring (*p* = 0.214).

None of the neonates experienced birth asphyxia or required naloxone treatment, resuscitation, or admission to the NICU. However, tachypnea was the cause of neonatal nursery admission. Approximately 10% of the postpartum women in each group reported difficulty in initiating breastfeeding. Eighty percent of participants in the 25 µg fentanyl group and almost 80% in the 50 µg fentanyl group were satisfied with their pain relief treatment (Table 5).

## 4. Discussion

This study aimed to identify appropriate non-axial opioid analgesia for relieving labor pain during the active phase based on the new definition of this phase [22,30], focusing on its implementation in low-resource settings. Short-acting opioids such as fentanyl, which have minimal adverse effects on the mother and fetus, are the drugs of choice. In addition, the CDC Clinical Practice Guidelines for Prescribing Opioids recommend prescribing the lowest effective dose of opioids to minimize adverse effects [20]. Although the effective dose of fentanyl can be determined through guidelines, some studies have shown that a 25 µg dose of fentanyl is also effective [15,18].

A key strength of this study is that it is the first trial to evaluate and compare the effectiveness of 25 µg fentanyl with the standard 50 µg dose during the active phase of labor, based on the new definition [22,30]. This study was conducted in accordance with good clinical practice for a double-blind randomized controlled trial.

This study demonstrated that both the 25 µg and 50 µg intravenous fentanyl regimens were effective in relieving labor pain during the active phase. At 30 min, pain scores decreased by 1.57 cm in the 25 µg group and 1.69 cm in the 50 µg group, consistent with the previously reported reduction of 0.9–1.6 cm [13,16,17,28]. However, unlike earlier studies [13,14,15,16], analgesic effects in our trial did not persist beyond 60 min. This discrepancy may be attributable to differences in timing of administration, as our protocol initiated treatment at ≥5 cm cervical dilation, whereas other studies began at 3–4 cm. Evidence indicates that providing analgesics at the earliest onset of pain enhances efficacy [31]. Furthermore, the waning effect observed after 60 min aligns with fentanyl’s known duration of action (30–60 min). Thus, earlier initiation and shorter redosing intervals may help maintain more sustained pain relief.

Additional analyses addressed concerns regarding body weight and weight-normalized dosing. No outliers were identified, and body weight and BMI distributions were comparable between groups. The recalculated cumulative fentanyl doses corresponded to 0.49 µg/kg in the 25 µg group and 0.87 µg/kg in the 50 µg group, values that approximate the recommended starting dose of 0.5 µg/kg. Although more participants in the 25 µg group required additional protocol doses, rescue doses were observed only in the 50 µg group. These findings suggest that the fixed 1 h dosing interval may have been longer than optimal for maintaining consistent pain relief.

Interestingly, a paradoxical trend was observed in which the 25 µg group reported slightly lower VAS scores despite receiving a lower absolute dose. This likely reflects baseline differences and individual variability in pain perception. Importantly, both regimens were safe, provided meaningful pain reduction at 30 min, and remained well below the protocol’s maximum cumulative dose of 500 µg. Notably, the 25 µg regimen achieved comparable pain relief while requiring significantly less fentanyl overall, supporting its consideration as a viable option for labor analgesia.

No serious neonatal adverse events occurred; however, approximately one-third of the neonates in both groups demonstrated abnormal intrapartum cardiotocography traces, and approximately 10% required nursery admission. Approximately 80% of the participants in both groups were satisfied with pain relief, while approximately 10% encountered difficulty initiating breastfeeding.

Maternal and neonatal safety outcomes in this study were similar to those reported in other studies [13,15,16,17,19]. Owing to local concerns regarding the new protocol for labor pain relief using fentanyl, continuous intrapartum cardiotocography monitoring was performed for 20 min in all the participants. Abnormal fetal monitoring was identified in approximately 35% of participants in each group, which is consistent with the findings of previous studies [13,19]. However, four participants in each group did not respond to conservative management for more than 40 min, leading to expedited cesarean deliveries. Only one neonate required nursery admission. Some studies have considered these abnormal events to be idiosyncratic and found that fentanyl doses below 100 µg did not affect fetal cardiovascular or acid–base balance [32].

The duration of the active phase of labor in this study was shorter than that observed during spontaneous labor [33]. This finding is similar to the duration reported by Shoorab et al., where fentanyl was suggested to have shortened the active phase of labor [15]. However, Atkinson et al. reported that the uterine contraction pattern remained unchanged 1 h after fentanyl injection [28]. Østborg et al. found that the mean duration of the active phase of spontaneous labor, starting at 4 cm of cervical dilation, was 368 min for nulliparous women and 165 min for multiparous women [33]. In comparison, the median durations in this study were 185 min and 121 min in nulliparous and multiparous women, respectively. One possible explanation for this difference is that, as Zhang et al. revealed, the progression of labor from 4 to 6 cm was longer than previously defined [34].

Although the pain score reduction was approximately 1–2 cm, approximately 80% of the participants reported satisfaction with the fentanyl regimen for pain relief. This finding is consistent with two previous experimental studies [16,17]. This satisfaction reflects participants’ perception of meaningful analgesic benefit without adverse effects from fentanyl, rather than satisfaction with the outcome of childbirth itself. Additionally, positive emotional involvement after giving birth may have contributed to this overall perception of treatment satisfaction [35].

Approximately 10% of the postpartum women in each group encountered difficulty establishing breastfeeding, which is consistent with previous studies [17,18]. Oommen et al. reported that fentanyl has a negative effect on the spontaneous suckling of newborn babies (odds ratio, OR 2.18; 95%CI 1.44–3.29) [18]. However, this study found a lower rate of early breastfeeding issues (10 vs. 32%), likely because the total fentanyl dose was less than half of that used in the study by Oommen et al. These findings support the idea that the fentanyl dose may influence early breastfeeding difficulties [18].

Statistical power was used to estimate the sample size for this study, specifically to assess the primary objective of comparing pain score reductions 30 min after treatment with 25 µg and 50 µg intravenous fentanyl doses. Unfortunately, approximately 25% and 50% of data were missing at the 60 and 120 min time points, respectively. The missing data were due to the trial being conducted during the active phase of labor, including nulliparous and multiparous women who delivered before the scheduled observation times. Consequently, the small sample size became a limitation, reducing the ability to assess repeated measurement outcomes. Additionally, this study did not include a sensitivity or subgroup analysis of factors that could contribute to variations in labor pain, such as parity. Furthermore, no long-term follow-up data was available.

## 5. Conclusions

To summarize, the effectiveness of pain relief during the active phase of labor after 30 min of treatment with 25 µg intravenous fentanyl was comparable to that of 50 µg intravenous fentanyl. Both regimens were safe for maternal use and resulted in high patient satisfaction. Although serious neonatal adverse events did not occur, concerns remain regarding abnormal intrapartum fetal monitoring and challenges with breastfeeding initiation. These findings can help inform pregnant individuals and their companions, and guide healthcare providers in delivering appropriate intrapartum care. Appraising the comparable effectiveness in relieving labor pain and the satisfaction with both 25 µg and 50 µg doses, along with the benefit of using a lower amount of the drug, this study suggests that using 25 µg intravenous fentanyl can be an alternative regimen for labor pain relief. However, further research is needed to determine the optimal time interval for fentanyl administration to ensure effectiveness, safety, and strategies to minimize fentanyl-related breastfeeding issues.

## Figures and Tables

**Figure 1 healthcare-13-02236-f001:**
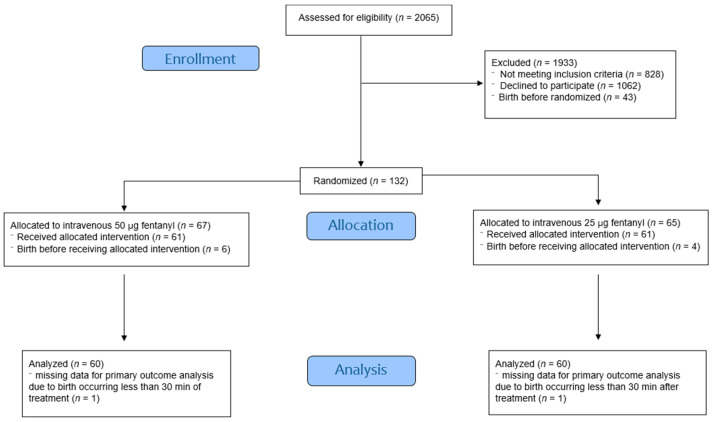
Study flow diagram according to CONSORT 2010 guidelines.

**Figure 2 healthcare-13-02236-f002:**
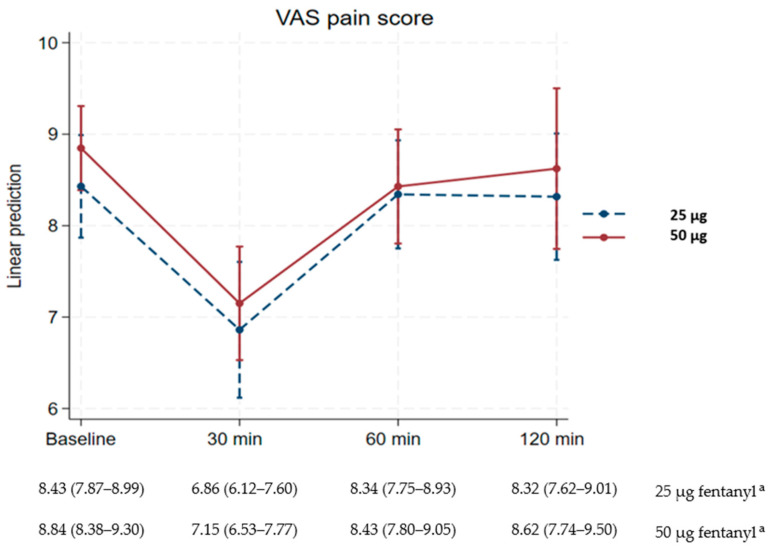
Visual analog scale (VAS) scores between 25 µg and 50 µg fentanyl regimens. ^a^ Data are reported as the margin of visual analog scale scores (95% confidence interval).

**Table 1 healthcare-13-02236-t001:** Baseline characteristics of participants by fentanyl group.

Characteristics	Category	Fentanyl 25 µg (*n* = 61)	Fentanyl 50 µg (*n* = 61)	*p*-Value
Age (years) ^a^	—	26.8 (23.5–28.5)	24.7 (22.6–29.5)	0.682
Marital status ^b^	Married	56 (91.8)	54 (88.5)	0.729
	Unspecified	5 (8.2)	7 (11.5)	
Education ^b^	Under bachelor	43 (70.5)	42 (68.9)	0.646
	Bachelor	18 (29.5)	19 (31.1)	
Body weight (kg) ^a^	—	65.5 (58.5–79.0)	65.0 (59.2–75.5)	0.697
BMI (kg/m^2^) ^a^	—	26.5 (23.5–31.2)	26.3 (23.7–29.7)	0.776
BMI group ^b^	18.5–24.9	26 (42.6)	25 (41.0)	0.700
	25–29.9	17 (27.9)	21 (34.4)	
	≥30	18 (29.5)	15 (24.6)	
Gestational age (weeks) ^c^	—	39.0 ± 0.9	38.9 ± 0.9	0.468
Parity ^b^	Nulliparous	35 (57.4)	33 (54.1)	0.592
	Multiparous	26 (42.6)	28 (45.9)	
Type of labor ^b^	Spontaneous	53 (86.9)	51 (83.6)	0.610
	Induction	8 (13.1)	10 (16.4)	
Cervical dilation at administration ^b^				0.908
	5–6 cm	51 (83.6)	50 (81.9)	
	7–8 cm	10 (16.4)	11 (18.1)	
Pregnancy complications ^b^	None	46 (75.4)	47 (77.1)	
	Hypertension	1 (1.6)	3 (4.9)	
	Gestational diabetes	3 (4.9)	5 (8.2)	
	Others	11 (18.1)	6 (9.8)	
Baseline VAS ^a^	—	8.9 (7.5–10.0)	9.2 (8.0–10.0)	0.255

^a^ Data are reported as median (interquartile range). ^b^ *n* (%). ^c^ Mean ± standard deviation.

**Table 2 healthcare-13-02236-t002:** Visual analog scale scores between fentanyl 25 µg and 50 µg regimens ^a^.

Time	Within-Group Change (25 µg) ^b^	*p*-Value	Within-Group Change (50 µg) ^b^	*p*-Value	Between-Group Difference ^c^	*p*-Value
Baseline ^d^	Ref.	–	Ref.	–	0.4 (−0.2, 1.1)	0.459
30 min ^e^	−1.57 (−2.1, −1.1)	<0.001	−1.69 (−2.2, −1.2)	<0.001	0.3 (−0.6, 1.2)	>0.999
60 min ^f^	−0.1 (−0.7, 0.5)	>0.999	−0.4 (−1.1, 0.3)	0.627	0.1 (−0.7, 0.9)	>0.999
120 min ^g^	−0.1 (−0.8, 0.6)	>0.999	−0.2 (−1.2, 0.7)	>0.999	0.3 (−0.7, 1.3)	>0.999

^a^ Multiple comparisons of visual analog scale (VAS) scores were performed using the Bonferroni correction method. Mean differences and mean changes were adjusted for the number of fentanyl doses administered. Data are reported as the mean difference (95% confidence interval). ^b^ Within-group analysis: pairwise comparisons of VAS scores at different time points in each fentanyl group. ^c^ Between-group analysis: comparisons of VAS scores between the fentanyl 25 µg and 50 µg groups. ^d^ Data analyzed from 122 participants (61 in each fentanyl group). ^e^ Data analyzed from 120 participants (60 in each fentanyl group). ^f^ Data analyzed from 88 participants (47 in the 25 µg group, 41 in the 50 µg group). ^g^ Data analyzed from 57 participants (33 in the 25 µg group, 24 in the 50 µg group).

**Table 3 healthcare-13-02236-t003:** Pain grade scales between 25 µg and 50 µg fentanyl regimens ^a^.

Pain Grade (VAS)	25 µg Baseline (*n* = 80) ^b^	25 µg 30 min (*n* = 75) ^c^	50 µg Baseline (*n* = 76) ^b^	50 µg 30 min (*n* = 71) ^c^
Mild (<4)	0	6 (8.0)	0	1 (1.4)
Moderate (≥4–6)	13 (16.2)	30 (40.0)	4 (5.3)	29 (40.9)
Severe (7–10)	67 (83.8)	39 (52.0)	72 (94.7)	41 (57.7)

^a^ *p*-value = 0.219, which is analyzed using Fisher’s exact test. ^b^ Baseline data include all protocols and additional and rescue doses (*n* = 80 in the 25 µg group; *n* = 76 in the 50 µg group). ^c^ Thirty-minute data exclude participants who delivered before the 30 min assessment (*n* = 75 in the 25 µg group; *n* = 71 in the 50 µg group).

**Table 4 healthcare-13-02236-t004:** Number of doses, rescue requirements, and total fentanyl used between intravenous 25 µg and 50 µg fentanyl regimens.

Outcomes	Fentanyl 25 µg (*n* = 61)	Fentanyl 50 µg (*n* = 61)	*p*-Value
Number of administered doses per protocol ^a,b^			0.201
One dose	44 (72.1)	52 (85.3)	
Two doses	15 (24.6)	8 (13.1)	
Three doses	2 (3.3)	1 (1.6)	
Rescue doses ^a^	0	5 (8.2)	0.057
Total fentanyl dose (µg) ^c^	32.8 ± 13.3	60.2 ± 22.1	<0.001
Total fentanyl dose (µg/kg) ^d,e^	0.49 (0.45–0.67)	0.87 (0.69–0.94)	<0.001
Duration between last dose and delivery (min) ^e^	103 (55–165)	73 (45–189)	0.570

^a^ Data are reported as *n* (%). ^b^ All participants received the first dose per protocol; additional or rescue doses were permitted (total 156 doses: 80 in 25 µg group, 76 in 50 µg group). Five rescue doses were administered to five participants (all in the 50 µg group). ^c^ Mean ± standard deviation. ^d^ Weight-normalized dose calculated as µg/kg (total fentanyl received ÷ body weight). ^e^ Median (interquartile range).

**Table 5 healthcare-13-02236-t005:** Secondary maternal and neonatal outcomes between the intravenous 25 µg and 50 µg fentanyl regimens.

Outcomes	Fentanyl 25 µg (*n* = 61)	Fentanyl 50 µg (*n* = 61)	*p*-Value
**Baseline safety parameters**			
Pulse rate (beats per minute) ^a^	86.75 ± 11.3	86.79 ± 11.3	0.987
Mean arterial pressure (mmHg) ^a^	85.73 ± 11.1	86.18 ± 12.8	0.838
Oxygen saturation (%) ^b^	99 (98–99)	99 (98–99)	0.423
Sedative scores = 2 ^c,d^	2 (3.3)	0 (0)	0.496
**Safety parameters at 30 min** ^e^			
Pulse rate (beats per minute) ^a^	85.5 ± 10.7	85.3 ± 10.3	0.915
Mean arterial pressure (mmHg) ^a^	87.5 ± 11.0	86.3 ± 12.7	0.574
Oxygen saturation (%) ^b^	98 (98–99)	98 (98–99)	0.357
Sedative scores = 2 ^c,d^	7 (11.5)	7 (11.5)	>0.999
**Labor and delivery outcomes**			
Duration of active phase in vaginal birth (min) ^b^	185 (78–260)	121 (79–249)	0.512
Birth mode ^c^			0.777
Vaginal delivery	43 (70.5)	39 (63.9)	
Vacuum extraction	3 (4.9)	3 (4.9)	
CD for abnormal fetal monitoring	4(6.6)	4 (6.6)	
Cesarean delivery for cephalopelvic disproportion	11 (18.0)	15 (24.6)	
Abnormal electronic fetal monitoring ^c^			0.214
Minimal variability	4 (6.6)	5 (8.2)	
Repetitive late deceleration	4 (6.6)	3 (4.9)	
Acute bradycardia	12 (19.7)	9 (14.7)	
Repetitive variable deceleration	0 (0)	5 (8.2)	
Prolonged deceleration	0 (0)	1 (1.6)	
**Neonatal outcomes**			
Apgar score ^b^			
At 1 min	8 (8–9)	8 (8–9)	0.563
At 5 min	9 (9–10)	9 (9–10)	0.405
Birthweight (g) ^a^	3150.9 ± 366.4	3042.5 ± 366.0	0.105
Nursery admission ^c,f^	8 (13.1)	6 (9.8)	0.226
Breastfeeding difficulties ^c,g^	6 (11.3)	5 (9.1)	0.838
**Maternal satisfaction with pain relief treatment** ^c^			0.588
Unsatisfied and not sure	12 (19.7)	14 (23)	
Agree and totally agree	49 (80.3)	47 (77)	

^a^ Data are reported as mean ± standard deviations (SD). ^b^ Median (interquartile range). ^c^ *n* (%). ^d^ Pasero sedative score 2 = occasionally drowsy and easily aroused. ^e^ Vital signs at 30 min reflect values following the initial fentanyl administration. ^f^ Nursery admission includes neonatal transfer for observation or treatment within 2 h after birth. ^g^ Data are calculated for 108 mothers and neonates (53 in the intravenous 25 µg fentanyl group and 55 in the intravenous 50 µg fentanyl group). Breastfeeding difficulty was defined as difficulty establishing feeding within 2 h. The cumulative fentanyl dose in both groups remained well below the study protocol’s maximum (500 µg). In the 25 µg group, the mean ± SD total dose was 32.8 ± 13.3 µg (range: 25–75 µg); in the 50 µg group, 60.2 ± 22.1 µg (range: 50–150 µg). No participant approached the maximum limit.

## Data Availability

The data will be available upon request due to privacy and ethical restrictions.

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
