# Peer review of "Comparison of Two Low-Dose Regimens of Intravenous Fentanyl for Pain Relief During Labor: A Double-Blind Randomized Controlled Trial"

_healthcare, 2025, doi:10.3390/healthcare13172236_

Round 1
Reviewer 1 Report
Comments and Suggestions for Authors
Veeraphol Srinil et al have conducted a study comparing the analgesic effect of 25 versus 50 mcg of fentanyl for pain relief during labour. At the outset, it must be admitted that we really lack an objective method of measuring pain, and pain is what the patient says she has. Pain perception is affected by a very large number of confounding factors that we do not even know. Thus, one is forced to assume that these factors operate in both groups at the same level. Fentanyl is among the most potent of analgesics in clinical use, and has the propensity to cause respiratory depression, a lower dose will therefore be preferred if it produces equal analgesia.
The methodology used is the gold standard in clinical research, double blind controlled trials, are the best design that we have and that has been used. The results have been presented in a simple way that most readers will understand and appreciate. However, one point strikes me, how come the VAS scores following 25 mcg are consistently lower than those in the 50 mcg group (see Fig.2)? Statistically the difference may not reach a significant level, due to the high scatter of values, but the scores are lower for the lower dose group, though this group also required a higher number of rescue doses.
This means that there is an uneven distribution of the threshold at which patients in either group ask for rescue, the reason for which is not clear. The data on BMI of patients were skewed (not normally distributed) or there are outliers, hence a median was used in place of mean. However, this makes it difficult to understand the cause of the anomaly in Fig 2. It may have been better if the dose was presented as mcg/kg body weight, since BMI does not give us an idea of what was the dose on the basis of body weight. While this way of presentation may not solve the riddle, it could have taken us closer to finding out why both doses had similar effects. May be the authors could add this information.
Instead of presneting BMI of the participants, it might have been better to present their weight. Analgesic effects and adverse effects are usually related to the dose per kg body weight and not to the BMI. Even now if you could make this change, I am sure it will help readers.
Reviewer 2 Report
Comments and Suggestions for Authors
Dear authors, you have done a good research work that is reflected in the results. The subject of study proposes a viable option for addressing pain in childbirth in a multitude of settings and situations where other techniques may be insufficient or inadequate. Below, a series of suggestions for improvement in the manuscript are presented. As for the sections of the methodology, it is recommended to follow the outline of the CONSORT standards that are cited, since there are sections that are not present, such as those related to "data collection", or that are included in other less appropriate sections.
Line 20. It should be specified that it is satisfaction with the fentanyl dose.
Line 38: Check reference 2 (Colomar et al.)
Line 38-39, citation 3: can be ignored.
Line 45-46: citation 5; Is there more current related evidence?
Line 48, 49: citation 6: that opioids do not produce active metabolites, the cited article speaks only of remifentanil. It is suggested to provide more evidence on the non-production of active metabolites.
Line 53: it is recommended to add limitations of use in obstetrics.
Line 60-61: it is suggested to expand the most commonly used guidelines; From what has been described, it seems that doses of 100mg every 5 minutes can be used.
Line 61-64: contrasts with what is described in lines 49-50, clarify.
Line 66: Does it mean childbirth instead of preganancy?
Line 73-74: Provide more information about the study environment, as it is only described as a tertiary level hospital
Line 75-76: It is implied that only women who arrived in the latent phase of labor were recruited, ignoring those who already arrived in the active phase of labor.
Line 84-Line 86: unifying inclusion criteria
Line 101-102: Is the exam routine or was the vaginal exam only performed because of its relationship with participation in the study? Was this aspect explained to the woman? Were the risks associated with frequent vaginal examinations taken into account?
Line 104-105: Were the syringes for the additional doses also blinded?
Line 109: At that same time, were you also evaluating the rest of the eligibility criteria?
Line 114-115: were the extra doses of the same concentration?, clarify.
Line 118: describe how the nurses who administered the doses that had already reached the maximum allowed were calculated. Were they offered other alternatives if they have already reached the maximum doses?
Line 131: specify that it is physiological saline
Line 129-138: this description can be included in the text of the "Intervention", it would not be necessary to specify how the mixture was made, only that the syringes contained the same amount of solution with different concentrations.
Line 148: specify that it is not about satisfaction with the results of the birth.
Line 152: Were the different types of resuscitation taken into account?
Line 152: it is suggested to provide means of Apgar scores in addition to values by range.
Line 162-163: provide more data on the studies on the basis of which the sample calculation was made.
Line 168-169: If you think it is appropriate, you can provide data from the analysis by protocol.
Line 191: the 122 women do not match those described in the flowchart. It is recommended that the flowchart be more closely aligned with the recommendations of the CONSORT guide.
Line 192-193: If this was a withdrawal criterion, describe it in the methodology.
Line 200: provide the "p" values of the basal variables
Line 212: Improve the appearance of the table
Line 236: Improve table, define well which measures the values that are exposed refer to.
Table 5. If the 50-dose Fentanyl was administered on average, 10 doses would be the maximum 500, this means that all women were administered the maximum dose; the same happens in fentanyl group 25. Clarify whether all women were given the maximum doses.
The discussion and conclusions are adequate, although it requires a more extensive review after correcting the previous issues.
Round 2
Reviewer 2 Report
Comments and Suggestions for Authors
Dear authors, the review work undoubtedly provides significant improvements. In my opinion, your work is suitable for a solid scientific publication. Greetings and congratulations.